



# Robust winter warming over Eurasia under stratospheric sulfate geoengineering – the role of stratospheric dynamics

Antara Banerjee[1,2], Amy H. Butler[2], Lorenzo M. Polvani[3], Alan Robock[4], Isla R. Simpson[5], Lantao Sun[6]

[1]Cooperative Institute for Research in Environmental Sciences, University of Colorado Boulder, Boulder, CO, USA
[2]National Oceanic and Atmospheric Administration, Chemical Sciences Laboratory, Boulder, CO, USA
[3]Department of Applied Physics and Applied Mathematics, Columbia University, New York, NY, USA
[4]Department of Environmental Sciences, Rutgers University, NJ, USA
[5]Climate and Global Dynamics Laboratory, National Center for Atmospheric Research, Boulder, CO, USA
[6]Department of Atmospheric Science, Colorado State University, Fort Collins, CO, USA

*Correspondence to*: Antara Banerjee (antara.banerjee@noaa.gov)

**Abstract.** It has been suggested that increased stratospheric sulfate aerosol loadings following large, low latitude volcanic eruptions can lead to wintertime warming over Eurasia through dynamical stratosphere-troposphere coupling. We here
investigate the proposed connection in the context of hypothetical future stratospheric sulfate geoengineering in the Geoengineering Large Ensemble simulations. In those geoengineering simulations, we find that stratospheric circulation anomalies that resemble the positive phase of the Northern Annular Mode in winter is a distinguishing climate response which is absent when increasing greenhouse gases alone are prescribed. This stratospheric dynamical response projects onto the positive phase of the North Atlantic Oscillation, leading to associated side-effects of this climate intervention strategy, such as
continental Eurasian warming and precipitation changes. Seasonality is a key signature of the dynamically-driven surface response. We find an opposite response of the North Atlantic Oscillation in summer, when no dynamical role of the stratosphere is expected. The robustness of the wintertime forced response stands in contrast to previously proposed volcanic responses.

## 1 Introduction

Mitigation of greenhouse gas (GHG) emissions remains of utmost importance in counteracting anthropogenic climate change.
However, given the challenges of meeting temperature targets such as 1.5 or 2°C above preindustrial under current commitments to the Paris Agreement (Rogelj et al., 2016), methods of climate intervention – or geoengineering – are increasingly gaining attention as potential means to supplement, albeit not replace, climate mitigation and adaptation strategies (National Research Council, 2015). Albedo modification, also known as solar radiation management (SRM), describes one set of approaches which propose cooling the planet by reflecting sunlight to space. Among these approaches, confidence is highest
in stratospheric sulfate injections resulting in a net negative radiative forcing and, consequently, a cooling of the planet (Crutzen, 2006; MacMartin et al., 2018) through the scattering effect of sulfate aerosols. Compelling observational evidence



for the global cooling effects of stratospheric sulfate aerosol is offered by large, low-latitude volcanic eruptions which, to some extent, provide a natural analog for sulfate geoengineering. For example, the widely observed eruption of Mt. Pinatubo in June 1991 injected around 18 Tg $SO_2$ into the stratosphere (Guo et al., 2004), and reduced global and annual average surface

temperatures by 0.5°C for two years following the eruption (Soden et al., 2002) while other large explosive eruptions of the past century reduced global average temperatures by 0.1-0.2°C (Robock and Mao, 1994). Key limitations of the analogy to geoengineering are the transient nature of volcanic perturbations compared to the hypothetically continuous deployment of sulfate geoengineering (Duan et al., 2019; Robock et al., 2008, 2013), and the fact that any proposed future sulfate geoengineering would be performed alongside increasing concentrations of atmospheric greenhouse gases.

Despite these limitations, volcanic eruptions might provide some insight into the potential side-effects of sulfate geoengineering, of which we remain poorly informed (Robock et al., 2013). Of interest in this study is the suggestion that low-latitude volcanic eruptions cause warming over the Northern Hemisphere (NH) continents, specifically Eurasia, in the one or two winters following the eruption (Fischer et al., 2007; Robock, 2002; Robock & Mao, 1992; Shindell et al., 2004; Zambri & Robock, 2016). The proposed mechanism for the surface warming essentially involves dynamical coupling of the

stratosphere and troposphere (Graf et al., 1993, 2007; Kodera, 1994; Robock and Mao, 1992). Following injection into the tropical lower stratosphere, sulfate aerosols locally warm the region primarily through absorption of longwave radiation. This leads to an increased equator-to-pole temperature gradient which, by thermal wind balance, strengthens the stratospheric polar vortex. The strengthened stratospheric westerlies are suggested to propagate downwards and project onto the positive phase of the North Atlantic Oscillation (NAO), which is associated with anomalous warmth over Eurasia.

However, the detectability of a volcanically forced response at the extratropical surface has recently been disputed by Polvani et al. (2019). Using ensembles of simulations to separate the effects of external forcing from internal variability, they concluded that the Eurasian winter warming following the 1991 eruption of Mt Pinatubo, which averaged around 1°C, was largely a result of internal variability. Polvani and Camargo (2020) reach the same conclusion for the 1883 Krakatau eruption, and suggest that a forcing much larger than the one from those (already large) eruptions would be needed to cause a

detectable surface warming in winter over Eurasia. This suggestion provides one motivation for our study: do continuous sulfate-forcing geoengineering scenarios force significant warming effects on the wintertime surface temperature over Eurasia through stratosphere-troposphere dynamical coupling? Indeed, a variety of mechanisms have been proposed whereby warmer temperatures in the tropical lower stratosphere, an increased meridional temperature gradient and strengthened stratospheric westerlies may impact the tropospheric circulation to produce a poleward shift of the jet stream and project onto the positive

phase of the NAO (Haigh et al., 2005; Kushner & Polvani, 2004; Simpson et al., 2009; Song & Robinson, 2004; Wittman et al., 2007).

The aim of this paper is to investigate the potential role of stratosphere-troposphere coupling in producing warm anomalies over Eurasia in wintertime in the Geoengineering Large Ensemble (GLENS) simulations (Tilmes et al., 2018a). In those simulations, sulfur dioxide ($SO_2$) is injected into the tropical and subtropical lower stratosphere in an attempt to stabilize

three surface temperature metrics: the global mean temperature, the interhemispheric temperature gradient and the equator-to-





pole temperature gradient, over the course of the 21$^{st}$ century. A robust strengthening of the stratospheric polar jets has indeed been found under sulfate geoengineering scenarios (Ferraro et al., 2015; Richter et al., 2018; Tilmes et al., 2018b), and the GLENS simulations do show warmer winters over several high latitude locations (Jiang et al., 2019). Our aim here is to link these two aspects of the geoengineering response. The contribution of stratospheric dynamics to a dampened seasonal cycle in

temperatures, in addition to seasonal insolation variations, has been suggested by Jiang et al. (2019). We here explore in more depth the existence of a stratosphere-troposphere dynamical coupling pathway, and quantify its relevance for near-surface patterns of variability, temperature and hydrology over the North Atlantic and Eurasia. As for the recent assessment of the undetectable forced response to volcanic eruptions (Polvani et al., 2019; Polvani and Camargo, 2020), the GLENS single-model ensemble approach allows for a clean determination of the forced response against a backdrop of internal variability.

## 2 Methods

### 2.1 Simulations

The simulations used here were performed with the Community Earth System Model version 1 (CESM1), containing atmosphere, ocean, sea ice and land components. We briefly describe the atmospheric component of the model, the Whole Atmosphere Community Climate Model (WACCM) (Mills et al., 2017). The WACCM resolution is 0.9° (latitude) by 1.25°

(longitude) with 70 vertical levels up to a model top of 140 km. The model comprehensively represents stratospheric processes. The inclusion of interactive chemistry is one key improvement upon previous generations of models used to study sulfate geoengineering (e.g., Ferraro et al., 2015). Feedbacks from changing ozone concentrations have significant effects on the large-scale stratospheric circulation, especially on the Quasi-Biennial Oscillation (QBO), in the geoengineering scenario of this study (Richter et al., 2017), and it has been shown that these feedbacks can considerably reduce the midlatitude jet shift response to

increased $CO_2$ (Chiodo and Polvani, 2017, 2019). Interactive chemistry is also important for sulfate aerosol concentrations, which are prognostically determined by oxidation of $SO_2$ by OH to $H_2SO_4$ and subsequent microphysics within a modal aerosol scheme, the Modal Aerosol Module (MAM3) (Mills et al., 2017). The simulated perturbation to radiative forcing following the eruption of Mt. Pinatubo compares well to observed estimates, providing validation for the model's radiative effects of sulfate aerosol (Mills et al., 2017).

90       We analyze the GLENS simulations, which are fully described by Tilmes et al. (2018a). We consider three scenarios, all of which were performed under the Representative Concentration Pathway 8.5 (RCP8.5) emissions scenario for greenhouse gases: (i) Base: 20 ensemble members performed between 2010-2030; (ii) RCP8.5: 3 members of Base that were extended out to 2097; and (iii) GEO8.5: 20 ensemble members with added stratospheric geoengineering performed between 2020-2099, which were branched off from the 20 Base members; this experiment is named "Geoengineering" in Tilmes et al. (2018a) and

"GLENS" in a few other studies (e.g. Jiang et al., 2019; Simpson et al., 2019).

Geoengineering in the GEO8.5 runs is implemented as $SO_2$ injections at four locations (15°N and 15°S at 25 km and 30°N and 30°S at 22.8 km, at 180° longitude). A primary aim of the experimental design was to limit the side-effects of



geoengineering. To this end, the GEO8.5 simulations aimed to maintain three annual mean surface temperature metrics at 2020 levels - the global mean temperature, the inter-hemispheric temperature gradient and the equator-to-pole temperature gradient - using a feedback algorithm that annually adjusted $SO_2$ injection amounts (Kravitz et al., 2017). By the end of the GEO8.5 simulations, the total $SO_2$ injection rate is 52 Tg yr$^{-1}$, which is around 5 times the one-time injection used for WACCM simulations of the Mt. Pinatubo eruption. The ensemble approach is a strength of the GLENS simulations, which allows us to separate the forced geoengineering response (given by ensemble means) from the noise due to internal variability (determined from the spread across ensemble members) (Deser et al., 2012).

To supplement GLENS, we analyze an additional set of simulations: the GEOHEAT_S runs described in Simpson et al. (2019), which we label GEOHEAT for simplicity. These aim to isolate the impact of stratospheric heating by the additional sulfate aerosols present in the GEO8.5 simulations from other factors such as the longwave effects of increasing GHGs and the shortwave effects of sulfate aerosols. The GEOHEAT runs were performed under Base conditions (average radiative forcing over 2010-2030), but with additional stratospheric heating rates derived from the last 20 years (2075-2095) of the GEO8.5 simulations. The GEOHEAT ensemble contains 4 members; each year of this simulation is in fact a spin-up run, initialized from 1 January of each year of the first 4 Base simulations, and these one-year runs are combined to give a 20-year length for each member. We use these runs in preference to other continuously forced runs in Simpson et al. (2019) since the short lengths of the spin-up runs limit, by design, the surface warming from increasing stratospheric water vapor (Richter et al., 2017; Tilmes et al., 2018b). This would otherwise be a confounding influence on the surface climate responses of interest in this study, which is not present in the corresponding GEO8.5 simulations.

## 2.2 Trends and indices of dynamical variability

We compare trends within the RCP8.5, GEO8.5 and GEOHEAT simulations in order to apply regression methods which elucidate the coupling between stratospheric and tropospheric climate responses. Linear trends are appropriate since the climate responses under sulfate geoengineering analyzed here are approximately linear in time, and in $SO_2$ injection rate, as noted in previous studies for temperature (global mean) and precipitation (Simpson et al., 2019; Tilmes et al., 2018a). There are some exceptions such as drying over the Mediterranean in winter which mostly occurs later in the simulations (Simpson et al., 2019). Fields are seasonally averaged before trends are computed. We primarily analyze the NH wintertime (December-February; DJF) and, for comparison, the NH summertime (July-August; JJA). Trends within RCP8.5 and GEO8.5 are taken between 2020 and 2095. For GEOHEAT, its difference with the Base climatology gives the 65-yr response between average 2020 and 2085 conditions. An equivalent trend is calculated by dividing this response by 65 for comparison with RCP8.5 and GEO8.5 trends. In all cases, trends are shown per 30 years.

We compute two indices of NH variability:

1) The Northern Annular Mode (NAM) to quantify stratosphere-troposphere coupling. We adopt its common definition as the leading Empirical Orthogonal Function (EOF) of geopotential height anomalies (Baldwin and Thompson, 2009; Gerber et al., 2010). The EOF is computed within Base, with concatenated geopotential height fields (20 years in 20





runs) in order to give the best available representation of model variability. The calculation is then performed independently at each pressure level as follows: the global mean is removed, the monthly mean climatology is removed, the seasonal average is taken over DJF, and the EOF pattern is calculated over the region 20°-90°N using a square root of cos(latitude) weighting. We next perform a projection onto the leading EOF in each GEO8.5 simulation. Again, the global mean is removed, the Base monthly mean climatology is removed and the seasonal average is taken over DJF. The Principal Component (PC) timeseries is calculated by projecting these anomalies onto the leading EOF and standardizing with the corresponding Base PC standard deviation. We select the NAM at 50 hPa ($NAM_{50}$), which is a common lower stratospheric metric for investigating stratosphere-troposphere coupling.

2) The North Atlantic Oscillation (NAO) to diagnose atmospheric circulation changes over the North Atlantic. The NAO index is computed over the North Atlantic region of 20°-80°N, 90°W-40°E in two ways: a) from the leading EOF of sea level pressure (SLP) anomalies to show its surface behavior and b) from the leading EOF of zonal mean zonal wind anomalies at each pressure level over the North Atlantic region to diagnose stratosphere-troposphere coupling. Other details of the calculation are the same as for the NAM, with the exception that the global mean is not removed from the raw fields.

# 3 Results

## 3.1 Surface air temperature and precipitation responses

The main goal of this paper is to explain the forced wintertime warming trends over Eurasia, Greenland and the North Atlantic which are simulated under sulfate geoengineering over the period between 2020-2095, as shown in Fig. 1a. That winter warming is found despite stabilization of the annual-mean equator-to-pole temperature gradient by the feedback control algorithm, and contributes to a dampened seasonal cycle in temperature (Jiang et al., 2019). Continental warming is largely absent in summer (Fig. 1b) - this seasonality by itself suggests a dynamical influence of the stratosphere in winter. Furthermore, we show in Fig. 1d a distinct dipole response in Eurasian precipitation under geoengineering. There is statistically significant drying over the Mediterranean and southern Europe, and wetting to the north over Scandinavia and above 50°N in the North Atlantic (Fig. 1d; see also Simpson et al. (2019)). Precipitation trends in summer are reversed over Eurasia relative to winter (Fig. 1e), which is again suggestive of a stratospheric dynamical influence in winter.

To place these geoengineering responses into the broader context of other anthropogenic forcings, we emphasize that the magnitude of the winter warming in the GEO8.5 simulations amounts to approximately one third of the forced warming under RCP8.5 (Fig. 1c). This makes clear that a feedback control algorithm that only maintains large scale, zonal mean and annual mean temperatures, does not successfully alleviate local changes in surface temperature, such as this, which is important since it occurs in a populated area. As for precipitation, trends under geoengineering are largely a cancellation, or slight reversal, of wetting trends in winter over the northern latitudes under rising greenhouse gases in RCP8.5 (Fig. 1f).



It is also of interest to contrast the winter warming under the geoengineering scenario to the potential response of the midlatitude surface temperature to large, low latitude volcanic eruptions. Consider first the signal-to-noise ratio by the end-of-century in GEO8.5 (shown in Fig. 2a). Here, the signal is defined as the ensemble mean difference between GEO8.5 (2075-

165 2095 average) and Base (2010-2030 average), and the noise is defined as interannual variability (computed as the standard deviation of DJF annual averages in Base, across 20 years in 20 members). We find that the signal-to-noise ratio is relatively small compared to the large wintertime interannual variability over the northern extratropics, lying at just around or below 1 in northern Eurasia, and lower still in northeastern Eurasia (Fig. 2a). Nevertheless, the statistical significance of the *forced* response (ensemble mean relative to standard error in mean) is robust. We demonstrate this in Fig. 2b by showing the year at

170 which ensemble mean trends beginning in 2020 become statistically significant (and remain significant) at the 95% confidence level, i.e. with $X/\{\sigma/(\sqrt{(N-1)})\} > 2$ where $X$ is the ensemble mean trend, $\sigma$ is the spread across ensemble members and $N$ is the number of ensemble members (20) (following Deser et al. 2012). We find that trends become significant around mid-century over Eurasia (Fig. 2b). The robustness of the forced response in the GEO8.5 simulations is also illustrated by the strong agreement across all 20 ensemble members for 2020-2095 trends, as shown in Supplementary Fig. 1 (the same is true for

175 precipitation; see Supplementary Fig. 2). Of course, for an individual realization, like we would observe in the real world, it would take much longer than the time scales shown in Fig. 2b to detect a significant trend. The robustness of the forced response to sulfate aerosol injections stands in stark contrast to the lack of a simulated forced response in the winter following the eruptions of Pinatubo and Krakatau (Polvani et al., 2019; Polvani and Camargo, 2020), and is a result of the sustained[1] and continuously increasing sulfate forcing in this scenario of geoengineering.

180 **3.2 Stratosphere-troposphere dynamical coupling**

We here argue for the existence of a dynamical stratosphere-troposphere coupling pathway under geoengineering in NH winter. We begin by showing ensemble mean DJF-average trends in zonal mean zonal wind for GEO8.5, GEOHEAT and RCP8.5 in Fig. 3. Trends in individual members are shown in Supplementary Fig. 3.

 The GEO8.5 simulations exhibit a forced strengthening of the NH wintertime polar vortex (up to 5 m s$^{-1}$ per 30 years

185 around 10 hPa) (Fig. 3a). This is consistent with a thermal wind balance response to tropical lower stratospheric heating (Tilmes et al., 2018a). Richter et al. (2018) found the same result for the first GEO8.5 ensemble member, and we here confirm the robustness of the forced response across 20 ensemble members, particularly below 10 hPa (Supplementary Fig. 3). Indeed, the lower stratospheric polar vortex response in GEOHEAT is broadly reproduced (Fig. 3b), as also discussed in Simpson et al. (2019), confirming the link between the tropical lower stratospheric warming and strengthening of the polar vortex.

190 In contrast, under climate change alone (RCP8.5), the most notable response is a strengthening of the upper flanks of the subtropical jets (Fig. 3c), which is a robustly simulated response to increased greenhouse gas radiative forcing (Lorenz and

---

[1] The sulfate injections in the GEO8.5 simulations are equivalent to several eruptions comparable to the 1991 Pinatubo eruption *per year*, and *continuously applied* for many decades.





DeWeaver, 2007; Manzini et al., 2014). We find no statistically significant trend in the NH stratospheric polar vortex (Fig. 3c). The lack of a polar vortex response is consistent with large intermodel spreads found by previous studies, with model responses differing in sign even under large greenhouse gas forcing scenarios such as RCP8.5 and 4xCO2 (Ayarzagüena et al., 2018, 2020; Manzini et al., 2014; Simpson et al., 2018). With the caveat of the single-model nature of our study, the stronger NH polar vortex under sulfate geoengineering is therefore a robust and key difference in the forced climate response compared to increasing greenhouse gases alone.

While Figs. 3a and b show a clear strengthening of the NH stratospheric polar vortex under geoengineering, the tropospheric response appears weaker. However, the zonal mean view masks zonal asymmetries in the troposphere. So, we now focus on the region of interest, the North Atlantic (20°-80°N, 90°W-40°E). Figure 4 shows ensemble mean trends in the NAO index, computed as the leading principal component of zonal wind as a function of height (see Methods, Section 2.2). When focusing on the NAO, what becomes apparent is a downward extension of the forced dynamical signal, i.e., positive ensemble mean NAO trends from the stratosphere to the troposphere under geoengineering in GEO8.5 (Fig. 4a). The connection of this response to tropical lower stratospheric heating is supported, once again, by the similar response in GEOHEAT (Fig. 4b). That this stratosphere-troposphere response is forced by sulfate geoengineering is further underscored by the negligible or negative trends found under RCP8.5 (Fig. 4c).

Since we are ultimately interested in the surface circulation response and the associated impacts, we show in Fig. 5 ensemble mean trends in SLP and the surface NAO timeseries (based on SLP; see Methods). Winter trends under geoengineering in GEO8.5 show a band of increasing SLP over Eurasia with a reduction over the North Pole (Fig. 5a); the general pattern is found in almost every ensemble member (Supplementary Fig. 4). These SLP trends project strongly onto the positive phase of the surface NAO index. We see that the index is generally positive and increases at a rate of 0.17±0.07 per 30 yrs (Fig. 5d). Consistently, the GEOHEAT ensemble mean response is significantly above zero at 0.55±0.27 (Fig. 5d, magenta bar; see Simpson et al. (2019) for further comparison of the zonal wind responses between GEOHEAT and GEO8.5 in the North Atlantic sector).

Once more, we highlight the unique feature of the wintertime geoengineering response as compared to the response in RCP8.5 where no simulated trend is seen (0.02±0.20 per 30 yrs) in the DJF NAO index (Fig. 5c, f). There is again also a distinct seasonality, with an opposite response in the summer geoengineering run, where the NAO index in fact decreases throughout the century with a trend of -0.28±0.07 per 30 yrs (Fig. 5b, e). In the absence of the stratospheric polar vortex, the JJA response is likely related to ocean circulation changes in this model (Fasullo et al., 2018), although its robustness across other models remains to be determined.

### 3.3 Quantifying the impact of stratospheric dynamics on surface climate responses

Having shown substantial evidence connecting the stratospheric, tropospheric and surface wintertime circulation responses under geoengineering, we complete our analysis by determining what fraction of the surface temperature and precipitation responses, which we are ultimately interested in, this dynamical coupling can explain. First, within the GEO8.5 simulations,





we regress on a gridcell-by-gridcell basis the DJF timeseries of each surface field against that of $NAM_{50}$ (following Thompson et al. (2000)). The regression coefficient, multiplied by the trend in $NAM_{50}$, yields the part of the surface climate trend that is congruent with $NAM_{50}$. The residual is then the $NAM_{50}$-congruent trend subtracted from the total trend in the surface field. Secondly, we investigate the GEOHEAT responses – under the proposed mechanism connecting tropical lower stratospheric heating, a strengthening of the NH polar vortex, downward dynamical coupling and circulation-driven climate changes, the

GEOHEAT and $NAM_{50}$-congruent responses should be the same. Indeed, we will show that they reasonably are.

        Performing this analysis for the surface circulation response, we find in Fig. 6a that $NAM_{50}$-congruent trends in SLP indicate a more positive shift of the NAO than the full trend (compare to Fig. 5a), as seen in the deeper low over the Arctic and the high shifted towards the North Atlantic. Consistently, the $NAM_{50}$-congruent temperature and precipitation trends both depict dynamically-driven responses to a positive NAO phase. There is warming over the Eurasian continent (Fig. 6d), which

explains most of the full trend (Fig. 1a) (upwards of 60%). The dipole response of drying over southern Europe and wetting over northern Europe (Fig. 6g) is similar in pattern, but larger in magnitude, than the full trend (Fig. 1d). Furthermore, the SLP pattern, northern Eurasian warming and the dipole response in precipitation over western Europe in GEOHEAT (Fig. 6b, e, h) is very similar to $NAM_{50}$-congruent trends within GEO8.5 (Fig. 6a, d, g). We therefore conclude that downward dynamical stratosphere-troposphere coupling, which is ultimately driven by tropical lower stratospheric heating from sulfate aerosols, is

the major driver of Eurasian winter warming, and associated changes in precipitation, in this particular scenario of sulfate geoengineering.

        Some large residuals from the $NAM_{50}$ regressions remain in the surface climate responses, which therefore cannot be explained by dynamical stratosphere-troposphere coupling. Residual SLP trends are large and oppositely signed to the $NAM_{50}$-congruent portion (compare Fig. 6a and c) and thus diminish the full response (Fig. 5a). There are residuals of around 1-2°C

per 30 yrs in temperature trends over the Barents-Kara sea, Greenland and the North Atlantic (Fig. 6f). The literature offers some explanation for these temperature residuals: annual-mean forced warming around Greenland has been linked to changes in the hydrological cycle over the North Atlantic and an acceleration of the Atlantic Meridional Overturning Circulation (AMOC), although this might be a model dependent feature (Fasullo et al., 2018). The warming over the Barents-Kara sea is associated with sea ice losses in mid-winter and spring (Jiang et al., 2019), but the feedbacks remain to be investigated. For

precipitation, the residual is a drying over northwestern Europe and a wetting to the south (Fig. 6i), which also diminishes the full response (Fig. 1d) compared to the $NAM_{50}$-congruent response alone (Fig. 6g). The circulation residual (Fig. 6c) could be contributing to the precipitation residual, and there is also a possible role of an overall weakening of storm track activity under the combined influence of geoengineering and increasing greenhouse gases (Simpson et al., 2019).

        Unlike for the North Atlantic and Eurasia, the stratospheric NAM cannot explain most of the climate responses to

geoengineering over the Pacific and North America. The residual from the $NAM_{50}$ regression analysis shows reductions in North Pacific SLP (Fig. 6c), which combines with the opposing $NAM_{50}$-congruent portion (Fig. 6a) to cause a dipole pattern in the full GEO8.5 trend (Fig. 5a). There are also residual warming trends over North America (Fig. 6f) and a dipole response





in precipitation over the North Pacific (Fig. 6i) that mostly explain the full trends (Fig. 1a, d). The Pacific climate response in GLENS and under sulfate geoengineering in general merits future study.

260        We return to the main point that stratospheric dynamics are a key influence on the Eurasian surface climate in this scenario of geoengineering with a prominent seasonal wintertime signature. Without its influence, the residual from the $NAM_{50}$ regression suggests a negative NAO response under sulfate geoengineering in this particular model (Fig. 6b). Indeed, in the absence of the polar vortex in the NH summer, we have instead revealed a shift towards the opposite NAO phase (Fig. 5b, e).

## 4 Conclusions

We have investigated the role of stratospheric dynamics for Northern Hemisphere regional climate changes under continuous and steadily increasing stratospheric sulfate injections to meet multiple annual-mean surface temperature targets under the RCP8.5 scenario in the Geoengineering Large Ensemble (GLENS) simulations. This geoengineering approach avoids many of the large surface climate impacts of greenhouse gas forcing under RCP8.5, for example in temperature and precipitation, but there are residual impacts on the high latitude, wintertime NH which we have studied here. Sulfate aerosol-driven warming

of the tropical lower stratosphere and consequent strengthening of the stratospheric polar vortex is a key difference in the climate response under geoengineering compared to a non-geoengineered climate in this model, and adds to the robustness of this finding in previous studies. The strengthening NH polar vortex, as reflected in the stratospheric Northern Annular Mode, correlates well in time with forced surface climate responses: regression analysis suggests that an increasing NAM at 50 hPa leads to a positive trend in the wintertime North Atlantic Oscillation, and is consequently the main cause of Eurasian

continental winter warming and a dipole response in precipitation. Experiments forced with just stratospheric heating from aerosols further cement the major role of dynamical stratosphere-troposphere coupling over other effects such as seasonal changes in insolation as suggested by Kravitz et al. (2017), in leading to the NH wintertime surface climate changes (see also Simpson et al. (2019) and Jiang et al. (2019)). Trends in North Atlantic sea level pressure congruent with the stratospheric NAM are, however, offset by a negatively signed component that also dominates in the summer season; the causes of this need

to be investigated further. Since these are results from a single model and a unique geoengineering strategy, the robustness of the dynamically driven winter warming simulated here needs to be ascertained from other models and other sulfate injection strategies. A recent study also finds winter warming, and argues similarly for stratosphere-troposphere coupling, under the sulfur injection scenario (G6sulfur) of the GeoMIP experiments with the UKESM1 and CESM2-WACCM6 models (Jones et al., 2020).

285        The role of volcanic forcing in causing the observed wintertime warming following the large Pinatubo and Krakatau eruptions has recently been questioned in the face of large internal variability (Polvani et al., 2019; Polvani & Camargo, 2020). The findings of this paper add further evidence to those studies. The forced warming in the GLENS simulations by the end of the century, when sulfate emissions have reached around 50 $Tg(SO_2)$ $yr^{-1}$, equivalent to several eruptions like Mt. Pinatubo each year and sustained for many decades, is found to be within the $1\sigma$ spread of unforced extratropical wintertime interannual



variability, suggesting that a single large eruption is very unlikely to be detectable. Nonetheless, the forced response in the GLENS simulations is robustly demonstrated with the ensemble approach. The side-effects of smaller and perhaps more plausible sulfate injections on the Eurasian continent would be smaller than shown here, with longer timescales needed for detection, underscoring the need for mitigation of greenhouse gas emissions.

**Code and data availability**

The code used to perform this analysis is available at: https://github.com/antara-banerjee/glens. The Geoengineering Large Ensemble data are available via the Earth System Grid: www.cesm.ucar.edu/projects/community-projects/GLENS/ (doi:/10.5065/D6JH3JXX).

**Competing interests**

The authors declare that they have no conflict of interest.

**Author contributions**

A. Banerjee and A. Butler performed the analysis. A. Banerjee prepared the manuscript with contributions from all authors.

**Acknowledgements**

Antara Banerjee is supported by a University of Colorado Boulder CIRES Visiting Fellow Program funded by NOAA agreement no. NA17OAR4320101. Alan Robock is supported by NSF grants AGS-1617844 and AGS-2017113.

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






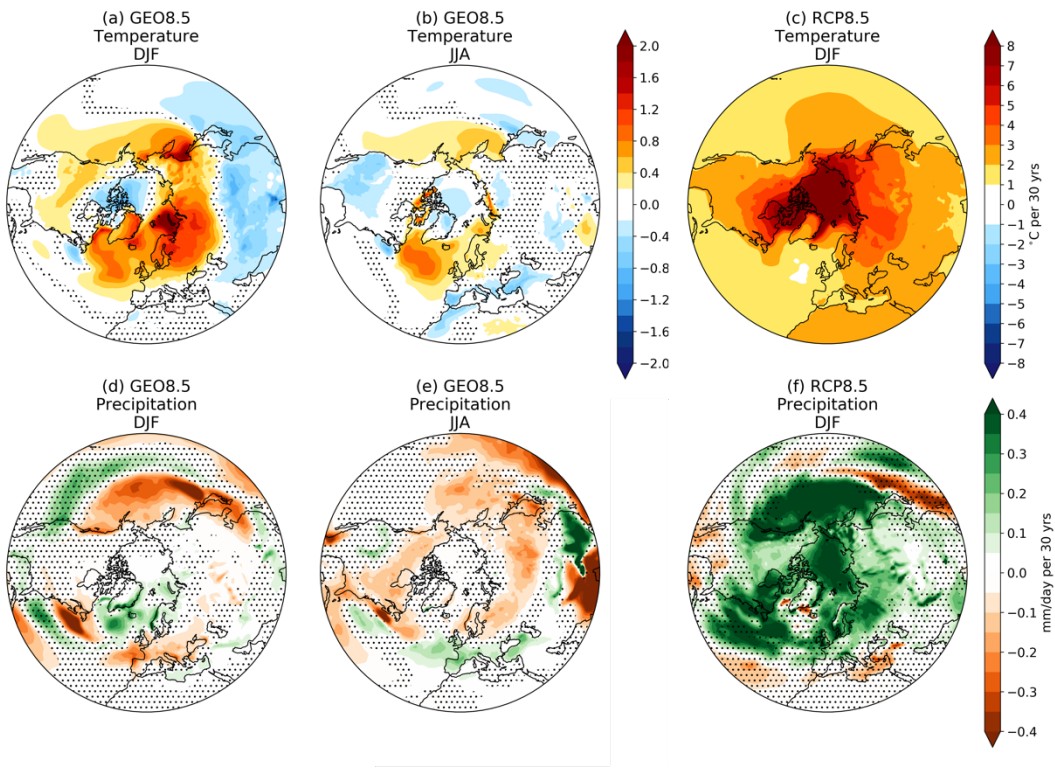

**Figure 1. Trends in near-surface air temperature and precipitation. Ensemble mean linear trends in (a, b, c) temperature (°C per 30 yrs) and (d, e, f) precipitation (mm/day per 30 yrs) in different seasons and experiments: GEO8.5, DJF (first column), GEO8.5 JJA (second column) and RCP8.5, DJF (third column). Stippling indicates lack of statistical significance at the 95% confidence level under a one sample Student's *t*-test using the standard deviation across ensemble members.**






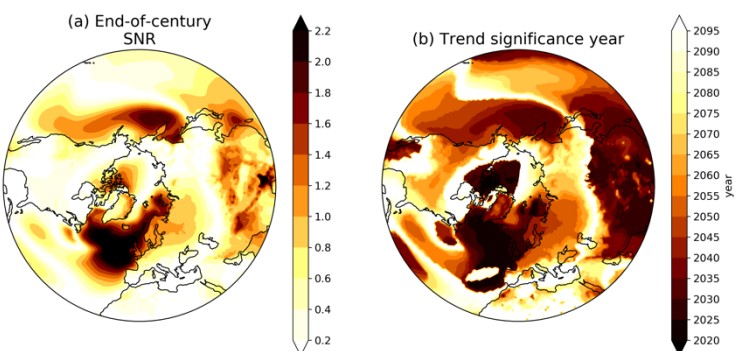

**Figure 2. Significance of the DJF near-surface air temperature response under geoengineering. (a) The signal to noise ratio of the end-of-century ensemble mean response in GEO8.5 (2075-2095) relative to Base (2010-2030). (b) The endpoint year at which GEO8.5 ensemble mean trends beginning in 2020 become statistically significant from zero, and remain significant, at the 95% confidence level.**





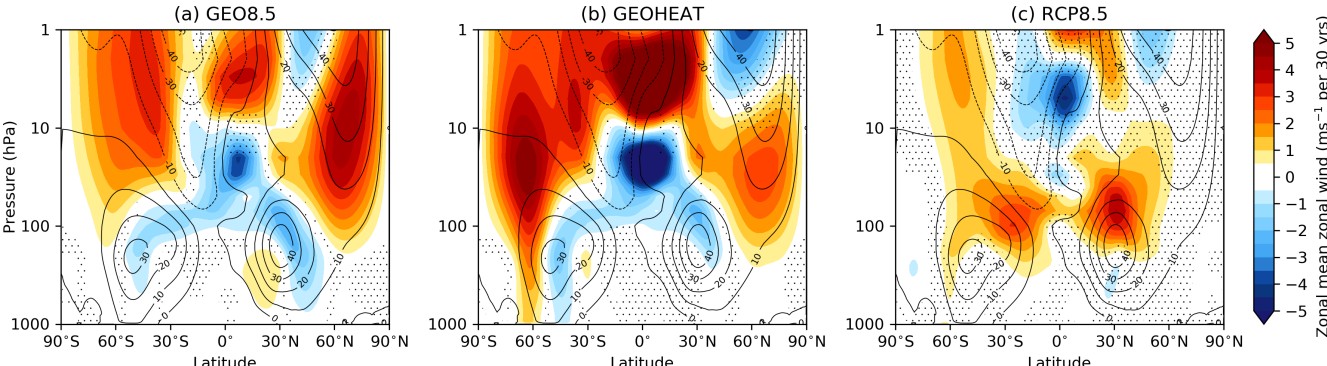

**Figure 3: Trends in DJF zonal mean zonal winds. Shown are ensemble mean, linear trends (shading; m s⁻¹ per 30 yrs) in (a) GEO8.5, (b) GEOHEAT and (c) RCP8.5. Contours show the Base climatology (2010-2030). Stippling indicates trends that are not statistically significant at the 95% confidence level under a one sample Student's *t*-test using the standard deviation across ensemble members.**





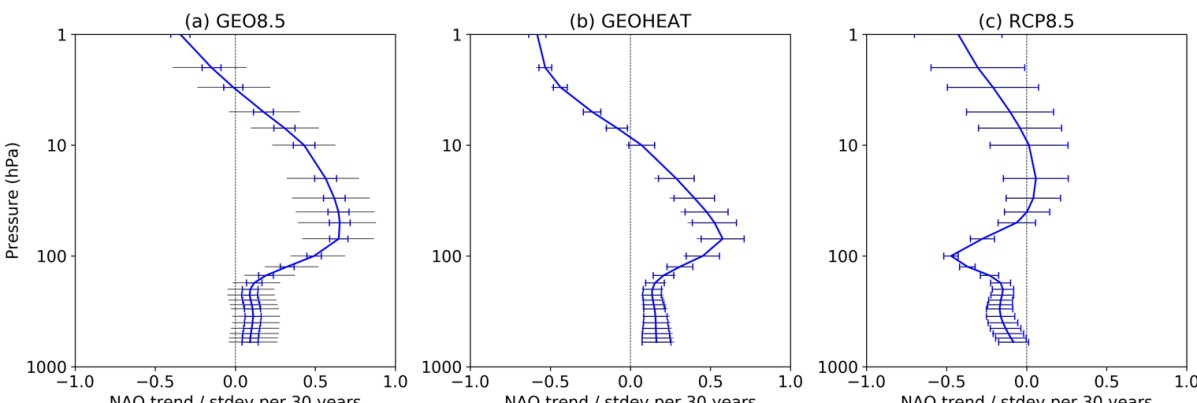

**Figure 4. Vertical profile of trends in DJF zonal wind characterizing the NAO. Shown are ensemble mean, linear trends in (a) GEO8.5, (b) GEOHEAT and (c) RCP8.5 with error bars showing the 95% confidence interval ($\pm 2\sigma/\sqrt{N}$ where N is the number of ensemble members). The 5th-95th percentile range is shown by horizontal black lines. In (b) GEOHEAT and (c) RCP8.5, the confidence interval and range are essentially the same.**





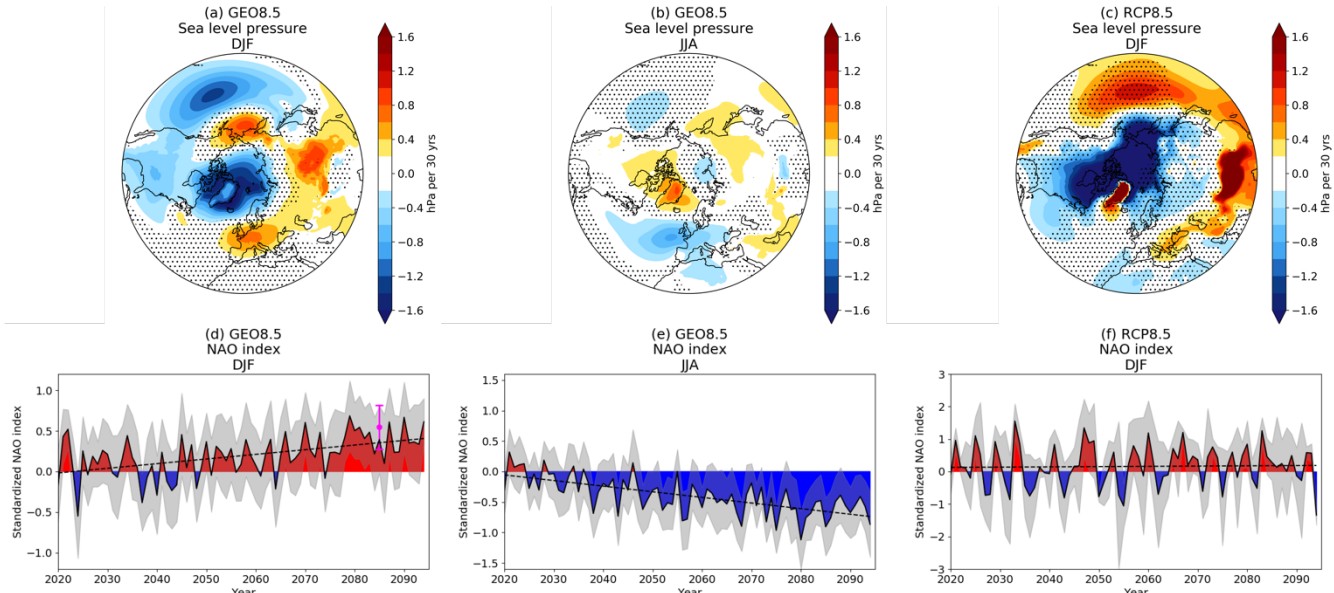


**Figure 5. Trends in SLP and the surface (SLP-based) NAO timeseries. (a, b, c) Ensemble mean linear trends in sea level pressure (hPa per 30 yrs). (d, e, f) Ensemble mean timeseries of the NAO. Responses are shown for GEO8.5, DJF (first column), GEO8.5, JJA (second column) and RCP8.5, DJF (third column). The magenta point in (d) shows the ensemble mean response in GEOHEAT and its 95% confidence interval. Stippling in (a, b, c) indicates lack of statistical significance at the 95% confidence interval under a one sample Student's *t*-test using the standard deviation across ensemble members. The grey shading in (d, e, f) indicates the 95% confidence interval ($\pm 2\sigma/\sqrt{N}$ where N is the number of ensemble members). The vertical scales for the NAO index are different in order to clearly illustrate the respective timeseries and confidence intervals.**


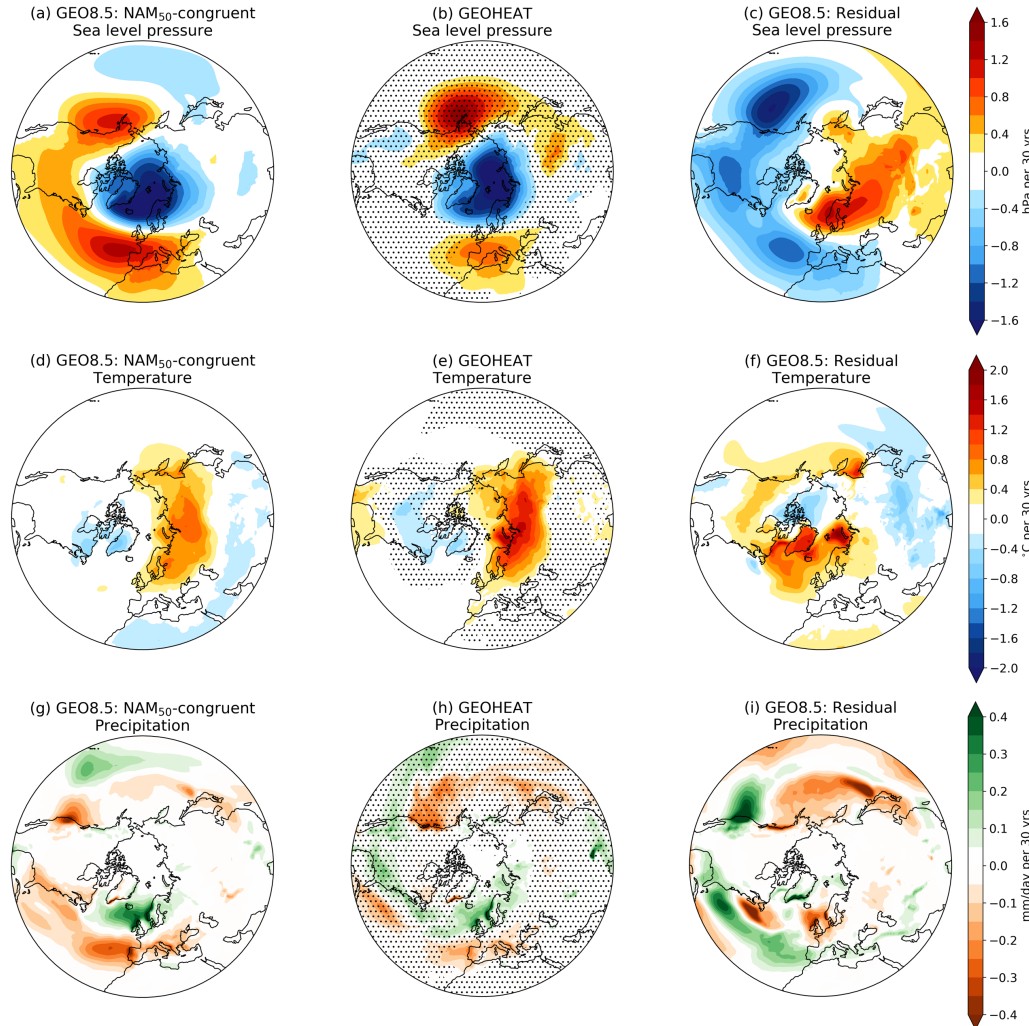

**Figure 6. Results from regressions of DJF surface climate responses against the stratospheric NAM in GEO8.5, and the responses in GEOHEAT. Ensemble mean linear trends in (a, b, c) sea level pressure (hPa per 30 yrs), (d, e, f) near-surface air temperature (°C per 30 yrs) and (g, h, i) precipitation (mm/day per 30 yrs). Shown are trends in GEO8.5 that are congruent with NAM$_{50}$ (first column), equivalent trends in GEOHEAT (second column), and the difference between total GEO8.5 trends and NAM$_{50}$-congruent trends (third column). In the second column, stippling indicates lack of statistical significance at the 95% confidence interval under a one sample Student's *t*-test using the standard deviation across ensemble members.**