# Peer review of "Robust winter warming over Eurasia under stratospheric sulfate geoengineering – the role of stratospheric dynamics"

_Atmospheric Chemistry and Physics, 2020_

## Referee Comment (RC1) · Anonymous Referee #1 · 15 Nov 2020

This is a useful, well done study that I think makes a valuable contribution to the literature. It addresses a question that, in my opinion, has not been handled terribly well in the past – how different are the dynamical effects of pulse versus sustained stratospheric sulfur injections? The paper is clearly motivated, well executed, and well written. I am recommending mostly minor comments. There's one set of comments that could require some work, but it hopefully isn't too onerous.

Lines 38-39: I'm not sure where the justification for these lines comes from.

Line 47: increased equator-to-pole temperature gradient aloft

Lines 50ff: As phrased, this somewhat undermines your point. You're saying that the

winter warming happens by chance, even for very large eruptions. So why would you expect to see a forced signal under geoengineering? I'd suggest reframing this paragraph.

Lines 158-159: Agreed with the sentence, but it's coming across as though this has not been recognized before.

Lines 159-160: I don't think the comment about this being in a populated area is compelling. There are lots of reasons why unpopulated areas might be important. The Antarctic ice sheet is a great example.

Line 167: Using an SNR threshold of 1 without context isn't all that helpful because the values are dependent on your definition. 1 might be a lot, or it might not be very much. It would be more useful to contextualize these numbers, discussing an area where you know that the answer is meaningful or not, and then use that to calibrate your SNR values. I find Figure 2b a lot more compelling than 2a in this regard, because it's a clear metric.

Line 200: I might suggest removing the latitude and longitude bounds. They imply that you're looking at changes in that box, which is very large and likely has a heterogeneous response.

Lines 225ff: I like this approach, but I think more details are needed and quite possibly an augmentation to your methodology. First, if there is reason to think that your DJF timeseries and NAM50 are both dependent variables (which seems likely), you will need to regress using an errors-in-variables method. Second, ordinary least squares regression doesn't inherently include complexities (like higher order terms) – this will show up as errors in your fits, possibly like what you find in the subsequent paragraph. If you have reason to believe that there are more complex relationships between the two variables, you can build that into your model.

Related to the previous comment, regressions have error bars. Is there a way to report
* * *
Interactive
comment

on that, perhaps with a figure that looks like the stippling that you show in Figure 6?

Lines 254ff: Some of your residuals resemble other known Pacific teleconnection patterns, at least at a glance. I agree that doing a good job with this is the subject of future work, but it might help if you could provide some candidate modes or mechanisms rather than being totally agnostic.

---

## Referee Comment (RC2) · Anonymous Referee #2 · 4 Dec 2020

The paper deals with the question of a possible winter warming in Eurasia as a consequence of artificial stratospheric sulfur injections. This winter warming has been proposed after large volcanic eruptions, but the scientific community has not been come to a final conclusion if the observed signal is really a consequence of the volcanic aerosols in the stratosphere. The numerical simulation of constant long lasting artificial injection of sulfur allows to get a better statistical evidence. The paper is well written, and contributes to a a still not well answered scientific question. The manuscript is within the scope of the journal and it clearly meets good scientific quality.

This study shows that a winter warming develops under solar radiation management

(SRM) when using sulfur injections. But the authors determine a very strong signal with a strong signal to noise ratio as the sulfur load is much stronger than after recently observed volcanic eruptions. The study needs to discuss this fact and should also discuss clearly the questions:

- May we expect a winter warming after a volcanic eruption? Can we conclude this from this study?

- Does the GLENS simulations with a long constant forcing help to answer this question? Forcing from SRM and volcanic forcing can be quite different. This is especially true for GLENS as the SO2 is injected at 15 N/S and 30 N/S. The resulting distribution of aerosols differs from a distribution after a volcanic eruption. Therefore, the gradient of the temperature anomaly will differ between the two cases. This needs to be described clearly in the article.

- How important is the signal to noise ratio? You determine a very strong signal. Sulfur load is much stronger than after recently observed volcanic eruptions. Which role plays the much stronger SRM forcing in the study compare to the Pinatubo forcing? GLENS is a transient simulation. You should add a comparison to the period when the SRM forcing is of comparable strength to a Pinatubo forcing.

Please, describe the proposed mechanism for the surface warming and stratosphere-troposphere coupling better in the introduction.

The article would gain from a figure of aerosol distribution and temperature anomaly and gradients in the stratosphere.

**Specific comments:**

Line 39: Why should SRM be performed alongside increasing GHG concentrations? This might be the most unwanted scenario.

Line 45: '...warm the region....' A few more words about the region, transport etc would be nice for the reader.

Line 47 'increased equator-to-pole gradient': Are you talking about the stratosphere?. Potential temperature has a minimum at the Equator. Warming of aerosols should decrease the gradient. Currently, end of November, we have -70C to -75C at 70 hPa in the tropics and and -70C to -73 C at the pole (https://earth.nullschool.net/). This is different at 10 hPa but the aerosol of SRM will mostly be at lower altitude, as well as volcanic sulfate.

Line 51: Which ensembles? Citation missing.

Line 58: Again, it is currently 10 to 15 K warmer at mid latitudes at 70 hPa than in the tropics and in at the Norther Pole.

Line 68: Warmer winters compared to control, RCP8.5 or?

Line 105: I have problems to understand the setup of the GEOHEAT experiment without reading Simpson et al (2019)

Line 149: Gradient in the stratosphere or troposphere? Do you meant avoiding the over cooling of the tropics with SRM (e.g. Kravitz et al, 2012)?

Line 159: '..such as this..' Where?

Line 170: How can the signal be significant over the Atlantic in 2020?

Line 179: The forcing of SRM in this scenario differs from a volcanic forcing. GLENS injects a four points. Does this impact the results? The transient GLENS simulation allows to determine winter warming in a period when the global forcing is similar to global Pinatubo forcing.

Line 220: Could the winter response also be an ocean or ice impact?

Line 229: Please. add a figure of stratospheric zonal mean temperature anomalies.

Line 266: Please, name the targets of GLENS again. This would be helpful for the reader.

Line 270: Warming is not only in tropical lower stratosphere in GLENS.

Figure 1: per 30 years? This is never described and the 30 years are never mentioned. Name the period and tell the reader what is the baseline.

Figure 3: Why do we see a strong positive trend in both hemispheres and why mainly in the summer hemisphere in GeoHEAT?

---

## Author Comment (AC1) · 18 Feb 2021

We are grateful to both reviewers for their very useful comments on this manuscript. Please find our point-by-point responses to the comments below in red font. Line numbers refer to the track changed manuscript.

**Response to Reviewer 1**

This is a useful, well done study that I think makes a valuable contribution to the literature. It addresses a question that, in my opinion, has not been handled terribly well in the past – how different are the dynamical effects of pulse versus sustained stratospheric sulfur injections? The paper is clearly motivated, well executed, and well written. I am recommending mostly minor comments. There's one set of comments that could require some work, but it hopefully isn't too onerous.

Lines 38-39: I'm not sure where the justification for these lines comes from.

We thank the reviewer for pointing out this line, which was misstated. It was our intention to state that geoengineering would likely be performed under higher background greenhouse gas concentrations than present at the time of historical volcanic analogs. The sentence has been corrected (L41-43).

Line 47: increased equator-to-pole temperature gradient aloft

**This sentence was removed in response to a comment from Reviewer #2.**

Lines 50ff: As phrased, this somewhat undermines your point. You're saying that the winter warming happens by chance, even for very large eruptions. So why would you expect to see a forced signal under geoengineering? I'd suggest reframing this paragraph.

Both the mentioned studies which question the volcanically forced winter warming (Polvani et al., 2019; Polvani and Carmago, 2020) do not deny the theoretical existence of the forcing – only that the forced response was likely overwhelmed by internal variability for the size of the eruptions considered. For example, Polvani et al. (2019) state "It is not impossible, we concede, that an extraordinarily large eruption (e.g., Tambora) may be capable of causing significant winter warming over the NH continents." These studies therefore do not undermine, but motivate, our geoengineering study: in the GLENS simulations, we have a scenario where sulfate injections are both continuous and very large (the SO2 injection rate in GEO8.5 is worth several Mt Pinatubo eruptions every year by the end of the simulation), and therefore offer the best hope for detecting the forced response above internal variability – as indeed we do. We have now explained this motivation more clearly on L72ff.

Lines 158-159: Agreed with the sentence, but it's coming across as though this has not been recognized before.

Agreed - we have added reference to previous studies which have recognized this for the GEO8.5 simulations (L184).

Lines 159-160: I don't think the comment about this being in a populated area is compelling. There are lots of reasons why unpopulated areas might be important. The Antarctic ice sheet is a great example.

We agree; we have removed the comment on population.

Line 167: Using an SNR threshold of 1 without context isn't all that helpful because the values are dependent on your definition. 1 might be a lot, or it might not be very much. It would be more useful to contextualize these numbers, discussing an area where you know that the answer is meaningful or not, and then use that to calibrate your SNR values. I find Figure 2b a lot more compelling than 2a in this regard, because it's a clear metric.

The reviewer's point here is well taken. Our intention, which we have now made clear (L191ff), is to place the SNR in the context of volcanic eruptions. An SNR value of 1, which means that the signal is  $1\sigma$  of interannual variability, would not be large if observed in the year or two following a volcanic eruption (e.g. see Figure 3 in Polvani and Carmargo (2020)). However, here, the robustness of the signal across all the members shows that it is a forced response to sulfate geoengineering.

Line 200: I might suggest removing the latitude and longitude bounds. They imply that you're looking at changes in that box, which is very large and likely has a heterogeneous response.

We have moved the latitude and longitude bounds to the next sentence.

Lines 225ff: I like this approach, but I think more details are needed and quite possibly an augmentation to your methodology. First, if there is reason to think that your DJF timeseries and NAM50 are both dependent variables (which seems likely), you will need to regress using an errors-in-variables method. Second, ordinary least squares regression doesn't inherently include complexities (like higher order terms) – this will show up as errors in your fits, possibly like what you find in the subsequent paragraph. If you have reason to believe that there are more complex relationships between the two variables, you can build that into your model.

Related to the previous comment, regressions have error bars. Is there a way to report on that, perhaps with a figure that looks like the stippling that you show in Figure 6?

We reply to these two comments together. We politely disagree that any more complex regression is necessary. We apply this widely used method (Thompson et al. 2000) as, admittedly, a simple look into the relationship between NAM50 and the DJF surface timeseries, with the acceptance that correlation in these two variables does not equal causality. The real complexity lies in our analysis of the GEOHEAT experiment. This experiment was carefully designed to isolate changes *caused by* aerosol-induced stratospheric heating, since that is the only perturbation in this experiment. It is the similarity between the NAM50-congruent and the GEOHEAT response which we emphasize as proving the role of stratospheric dynamical changes for surface climate changes. We now reiterate this point on L266.

We do agree, however, that we should show some measure of "error" in Figure 6. For best comparison with the other figures in the paper, we have added stippling showing 95% statistical significance calculated using the spread across ensemble members.

Lines 254ff: Some of your residuals resemble other known Pacific teleconnection patterns, at least at a glance. I agree that doing a good job with this is the subject of future work, but it might help if you could provide some candidate modes or mechanisms rather than being totally agnostic.

We have speculated on long term changes in ENSO in contributing to some of these residuals on L297ff.

**References**

Polvani, L. M., Banerjee, A. and Schmidt, A.: Northern Hemisphere continental winter warming following the 1991 Mt. Pinatubo eruption: reconciling models and observations, Atmos. Chem. Phys., 19(9), 6351–6366, doi:10.5194/acp-19-6351-2019, 2019.

Polvani, L. M. and Camargo S. J.: Scant evidence for a volcanically forced winter warming over Eurasia following the Krakatau eruption of August 1883, Atmos. Chem. Phys. Discuss., 1-25. doi: https://www.atmos-chem-phys-discuss.net/acp-2020-271/, 2020.

Thompson, D. W. J., Wallace, J. M. and Hegerl, G. C.: Annular Modes in the Extratropical Circulation. Part II: Trends, J. Climate, 13(5), 1018-1036, doi: 10.1175/1520-0442(2000)013<1018:AMITEC>2.0.CO;2, 2000.

**Response to Reviewer 2**

The paper deals with the question of a possible winter warming in Eurasia as a consequence of artificial stratospheric sulfur injections. This winter warming has been proposed after large volcanic eruptions, but the scientific community has not been come to a final conclusion if the observed signal is really a consequence of the volcanic aerosols in the stratosphere. The numerical simulation of constant long lasting artificial injection of sulfur allows to get a better statistical evidence. The paper is well written, and contributes to a still not well answered scientific question. The manuscript is within the scope of the journal and it clearly meets good scientific quality.

This study shows that a winter warming develops under solar radiation management (SRM) when using sulfur injections. But the authors determine a very strong signal with a strong signal to noise ratio as the sulfur load is much stronger than after recently observed volcanic eruptions. The study needs to discuss this fact and should also discuss clearly the questions:

• May we expect a winter warming after a volcanic eruption? Can we conclude this from this study?

Although we have attempted to draw parallels to volcanic eruptions, we are wary about using the sulfur injection strategy implemented in these simulations (that of continuous injections) to draw conclusions about the presence/lack thereof of winter warming from volcanic eruptions (which are abrupt injections). We have noted this key limitation of the analogy of sulfate SRM with volcanic eruptions in the Introduction (L37-39). The impacts of volcanic eruptions also depend on the nature of each individual eruption, such as its magnitude and latitude. Our intention, therefore, is not to make a judgement on whether winter warming after volcanic eruptions may be detectable - we leave this debate for dedicated studies on the subject (e.g. Polvani et al. 2019; Polvani and Camargo, 2020).

• Does the GLENS simulations with a long constant forcing help to answer this question? Forcing from SRM and volcanic forcing can be quite different. This is especially true for GLENS as the SO2 is injected at 15 N/S and 30 N/S. The resulting distribution of aerosols differs from a distribution after a volcanic eruption. Therefore, the gradient of the temperature anomaly will differ between the two cases. This needs to be described clearly in the article.

This is a good point: the idealized injection locations represent another limitation in the analogy of sulfate SRM with volcanic eruptions. We add this to our list of limitations in the Introduction (L39-41). It should be recognized that different volcanic eruptions will also show different aerosol distributions and temperature perturbations – we therefore draw the comparison to volcanic eruptions quite broadly.

• How important is the signal to noise ratio? You determine a very strong signal. Sulfur load is much stronger than after recently observed volcanic eruptions. Which role plays the much stronger SRM forcing in the study compare to the Pinatubo forcing? GLENS is a transient simulation. You should add a comparison to the period when the SRM forcing is of comparable strength to a Pinatubo forcing.

This study is not meant to be a test of whether the 1991 Mt. Pinatubo eruption was responsible for the observed winter warming in the winter of 1991-1992. A new paper by one of our authors (Coupe, Joshua, and Robock, Alan, 2021: The influence of stratospheric soot and sulfate aerosols on the Northern Hemisphere wintertime atmospheric circulation. Submitted to *Journal of Geophysical Research* – *Atmospheres.*) shows that it was, but that it depended on also having the correct sea surface

temperatures and tropospheric interactions. In the current paper, we show that large stratospheric forcing is sufficient to produce a positive AO mode, as others have shown before. Furthermore, because the Pinatubo forcing was episodic, with changing forcing as a function of time and space, there is no similar forcing in the GLENS runs we studied here, and certainly not cases where there were sea surface temperatures as observed after the Pinatubo eruption.

Please, describe the proposed mechanism for the surface warming and stratosphere-troposphere coupling better in the introduction.

There are many proposed mechanisms for how the stratosphere couples to the troposphere and it remains ambiguous which of these mechanisms fully explains the process. However, we have added references to a few of these theories on L55ff.

The article would gain from a figure of aerosol distribution and temperature anomaly and gradients in the stratosphere.

We add to Figure 3 the ensemble mean, zonal mean temperature trends in GEO8.5, GEOHEAT and RCP8.5 since these directly relate to the zonal wind changes of interest here. We also add a discussion of these temperature trends and 50 hPa temperature gradients to the beginning of Section 3.2. The aerosol distribution changes in GEO8.5 are comprehensively covered in numerous studies that we have cited (e.g. Tilmes et al. 2018a; Tilmes et al. 2018b).

**Specific comments:**

Line 39: Why should SRM be performed alongside increasing GHG concentrations? This might be the most unwanted scenario.

We thank both reviewers for pointing out this incorrect statement. We intended to state that geoengineering would likely be performed under higher background greenhouse gas concentrations than present at the time of historical volcanic analogs. The sentence has been corrected (L41-43).

Line 45: '...warm the region....' A few more words about the region, transport etc would be nice for the reader.

We have stated the dispersion and shortwave effects of sulfate aerosols in L50-53.

Line 47 'increased equator-to-pole gradient': Are you talking about the stratosphere?. Potential temperature has a minimum at the Equator. Warming of aerosols should decrease the gradient. Currently, end of November, we have -70C to -75C at 70 hPa in the tropics and and -70C to -73 C at the pole (https://earth.nullschool.net/). This is different at 10 hPa but the aerosol of SRM will mostly be at lower altitude, as well as volcanic sulfate. °

We meant here that the change in the gradient (temperature at equator minus temperature at pole) is positive. However, we take the reviewer's point that this is poorly phrased. In describing stratospheretroposphere coupling mechanisms, we have removed this sentence. We have also added a quantitative discussion of lower stratospheric (50 hPa) temperature gradient changes to the beginning of Section 3.2, and shown values in Fig. 3a-c.

Line 51: Which ensembles? Citation missing.

We have now noted on L69 that Polvani et al. (2019) used large ensembles of simulations from the CESM-WACCM, CESM-CAM5 and CanESM2 models.

Line 58: Again, it is currently 10 to 15 K warmer at mid latitudes at 70 hPa than in the tropics and in at the Norther Pole.

In moving these lines to the previous paragraph to discuss stratosphere-troposphere coupling mechanisms, we have removed "meridional temperature gradient".

Line 68: Warmer winters compared to control, RCP8.5 or?

We have clarified on L89 that the difference is relative to a baseline of 2010-2030 average temperatures under no artificial sulfate injections.

Line 105: I have problems to understand the setup of the GEOHEAT experiment without reading Simpson et al (2019)

Yes, the experimental design of GEOHEAT is somewhat arduous to explain. We have moved some of the experimental detail from Section 2.1, L132ff to footnote (1) and left the main paragraph with the essential information. We have also added a sentence at the end of the paragraph summarizing how and why we use GEOHEAT.

Line 149: Gradient in the stratosphere or troposphere? Do you meant avoiding the over cooling of the tropics with SRM (e.g. Kravitz et al, 2012)?

The GEO8.5 simulations control surface temperature gradients (as well as global mean surface temperature) in order to avoid tropical overcooling. We add in "surface" to L173.

Line 159: '.. such as this..' Where?

We meant the Eurasian winter warming trends in GEO8.5, but we have removed the phrase "such as this" to avoid confusion (L184).

Line 170: How can the signal be significant over the Atlantic in 2020?

The dark color over the Atlantic represents trends becoming significant between 2020 and 2025, not at 2020.

Line 179: The forcing of SRM in this scenario differs from a volcanic forcing. GLENS injects a four points. Does this impact the results? The transient GLENS simulation allows to determine winter warming in a period when the global forcing is similar to global Pinatubo forcing.

Please see our previous answer: we believe it is not appropriate to quantitatively compare the GLENS simulations to Pinatubo due to possible tropospheric interference from sea surface temperatures in the case of Pinatubo, as well as the different forcing in time and space (including the artificial four-point injection in GLENS as the reviewer points out).

Line 220: Could the winter response also be an ocean or ice impact?

Please refer to Section 3.3, paragraph beginning L281, where we have already discussed which features of the atmospheric winter climate response are more likely to be related to ocean/ice changes rather than dynamically induced by stratospheric changes, i.e. the residuals from the NAM-regression analysis in the final column of Fig. 6.

Line 229: Please. add a figure of stratospheric zonal mean temperature anomalies.

We have added subplots showing zonal mean temperature trends in GEO8.5, GEOHEAT and RCP8.5 to Fig. 3.

Line 266: Please, name the targets of GLENS again. This would be helpful for the reader.

We have now reiterated the GLENS temperature targets on L309-310.

Line 270: Warming is not only in tropical lower stratosphere in GLENS.

We only state that tropical lower stratospheric warming is  $\underline{a}$  key response (not the only response) to geoengineering.

Figure 1: per 30 years? This is never described and the 30 years are never mentioned. Name the period and tell the reader what is the baseline.

Trend calculations are carefully described in Section 2.2: "Trends and indices of dynamical variability". We state on L150 that "In all cases, trends are shown per 30 years". We are not sure what the reviewer means by 'baseline' here. For RCP8.5 and GEO8.5, the trends are in the individual experiments, not relative to Base. For GEOHEAT, the response is relative to Base and is converted to an equivalent trend. This is all explained in detail in Section 2.2.

Figure 3: Why do we see a strong positive trend in both hemispheres and why mainly in the summer hemisphere in GeoHEAT?

Firstly, the Northern Hemisphere vortex in GEOHEAT strengthens less than in GEO8.5 due to the smaller change in meridional temperature gradient in the former case (see our new discussion in Section 3.2). The polar vortex in both hemispheres is expected to strengthen in these simulations; in the Southern Hemisphere, we speculate that the GEO8.5 response could be smaller than in GEOHEAT due to the effects of ozone recovery in GEO8.5. Ultimately, how the polar vortex responds to stratospheric heating will depend on how a new balance involving the altered stratospheric heating, altered wave driving and meridional circulations is achieved, so we should not necessarily expect the same magnitude response in each hemisphere since the wave and polar vortex dynamics are different.

**References**

Polvani, L. M., Banerjee, A. and Schmidt, A.: Northern Hemisphere continental winter warming following the 1991 Mt. Pinatubo eruption: reconciling models and observations, Atmos. Chem. Phys., 19(9), 6351–6366, doi:10.5194/acp-19-6351-2019, 2019.

Polvani, L. M. and Camargo S. J.: Scant evidence for a volcanically forced winter warming over Eurasia following the Krakatau eruption of August 1883, Atmos. Chem. Phys. Discuss., 1-25. doi: https://www.atmos-chem-phys-discuss.net/acp-2020-271/, 2020.

Tilmes, S., Richter, J. H., Kravitz, B., MacMartin, D. G., Mills, M. J., Simpson, I. R., Glanville, A. S., Fasullo, J. T., Phillips, A. S., Lamarque, J.-F., Tribbia, J., Edwards, J., Mickelson, S. and Ghosh, S.: CESM1(WACCM) Stratospheric Aerosol Geoengineering Large Ensemble Project, Bull. Amer. Meteor. Soc., 99(11), 2361–2371, doi:10.1175/BAMS-D-17-0267.1, 2018a.

Tilmes, S., Richter, J. H., Mills, M. J., Kravitz, B., MacMartin, D. G., Garcia, R. R., Kinnison, D. E., Lamarque, J.-F., Tribbia, J. and Vitt, F.: Effects of Different Stratospheric SO2 Injection Altitudes on Stratospheric Chemistry and Dynamics, J. Geophys. Res. Atmos., 123(9), 4654–4673, doi:10.1002/2017JD028146, 2018b.